# Minimally invasive surgical techniques for oesophageal cancer and nutritional recovery: a prospective population-based cohort study

Joonas H Kauppila [1,2] Helen Rosenlund,[1,3] Fredrik Klevebro,[1] Asif Johar,[1] Poorna Anandavadivelan [1] Kalle Mälberg,[1] Pernilla Lagergren[1,4]

¹Surgical Care Sciences, Karolinska Institutet, Karolinska University Hospital, Stockholm, Sweden
²Department of Surgery, University of Oulu, Oulu University Hospital, Oulu, Finland
³Department of Orthopaedics, Danderyds Sjukhus AB, Stockholm, Sweden
⁴Department of Surgery & Cancer, Imperial College London, London, UK

**Correspondence to**
Prof Joonas H Kauppila;
joonas.kauppila@ki.se

## ABSTRACT

**Objectives** To explore whether the minimally invasive oesophagectomy (MIE) or hybrid minimally invasive oesophagectomy (HMIE) are associated with better nutritional status and less weight loss 1 year after surgery, compared with open oesophagectomy (OE).

**Design** Prospective cohort study.

**Setting** All patients undergoing oesophagectomy for cancer in Sweden during 2013–2018.

**Participants** A total of 424 patients alive at 1 year after surgery were eligible, and 281 completed the 1-year assessment. Of these, 239 had complete clinical data and were included in the analysis.

**Primary and secondary outcome measures** The primary outcome was nutritional status at 1 year after surgery, assessed using the abbreviated Patient-Generated Subjective Global Assessment questionnaire. The secondary outcomes included postoperative weight loss at 6 months and 1 year after surgery.

**Results** Of the included patients, 78 underwent MIE, 74 HMIE while 87 patients underwent OE. The MIE group had the highest prevalence of malnutrition (42% vs 22% after HMIE vs 25% after OE), reduced food intake (63% vs 45% after HMIE vs 39% after OE), symptoms reducing food intake (60% vs 45% after HMIE vs 60% after OE) and abnormal activities/function (45% vs 32% after HMIE vs 43% after OE). After adjustment for confounders, MIE was associated with a statistically significant increased risk of reduced food intake 1 year after surgery (OR 2.87, 95% CI 1.47 to 5.61), compared with OE. Other outcomes were not statistically significantly different between the groups. No statistically significant associations were observed between surgical techniques and weight loss up to 1 year after surgery.

**Conclusions** MIE was statistically significantly associated with reduced food intake 1 year after surgery. However, no differences were observed in weight loss between the surgical techniques. Further studies on nutritional impact of surgical techniques in oesophageal cancer are needed.

## STRENGTHS AND LIMITATIONS OF THIS STUDY

⇒ The strength of this study is the prospective and population-based study design with longitudinal collection of data according to a predefined study protocol.
⇒ The relatively high inclusion rate and complete data are strengths.
⇒ The abbreviated Patient-Generated Subjective Global Assessment questionnaire is validated and commonly used to measure malnutrition in oncology patients.
⇒ A weakness of this study was that all patients who died during 1 year after oesophagectomy were not eligible.

with preoperative neoadjuvant therapy.[2] The traditional open oesophagectomy (OO) with abdominal and thoracic incisions is being replaced by minimally invasive oesophagectomy (MIO), performed using thoracoscopy and laparoscopy.[3] MIE is associated with lower risk of postoperative complications, especially pulmonary, lower in-hospital mortality, shorter length of hospital stay and lower 90-day mortality, compared with OE.[4–7] MIE has also been associated with improved health-related quality of life (HRQoL).[8 9] Hybrid minimally invasive oesophagectomy (HMIO) applies open surgery in combination with laparoscopic or thoracoscopic technique, for example, a thoracotomy and laparoscopy. HMIE is associated less postoperative mortality and better HRQoL, compared with OE.[10]

Oesophageal cancer is associated with significant nutritional problems and weight loss before and after treatment.[11–13] Patients lose between 5% and 12% of their preoperative weight during the first 6 months after oesophagectomy and a majority have lost more than 10% of their body weight 12 months postoperatively.[14] However, the

## INTRODUCTION

The mainstay of curative treatment for oesophageal cancer is surgery which carries a high risk of postoperative complications and mortality.[1] The surgery is often combined

**BMJ**

effect of MIE on nutritional recovery remains unclear. It was hypothesised that MIE could improve nutritional recovery after oesophagectomy, compared with OE.

The aim of this study was to investigate the impact of surgical technique, used for oesophagectomy for cancer, on nutritional status and weight loss up to 1 year after surgery in a population-based and nationwide setting.

## MATERIALS AND METHODS
### Study design
This project was based on the nationwide and prospective cohort entitled Oesophageal Surgery on Cancer patients-Adaptation and Recovery (OSCAR).[15] Patients who underwent surgery for oesophageal cancer in Sweden from 1 January 2013 and onwards were included. Recruitment was done 1 year after surgery, and thus the inclusion started 1 January 2014. Eligible patients were identified from pathology units at eight treating hospitals in Sweden. An invitation letter was sent and within 1 week patients were contacted by telephone to get information regarding the project and to give their consent to participate. Arrangements for the first assessment were also made. A research nurse visited the patients in their homes for collection of patient-reported outcomes, objective measures of body weight and body composition.

Clinical data at time of surgery were collected from medical records, which were reviewed according to a predefined study protocol to ensure uniformity: (1) patient characteristics including age, sex, comorbidities and weight at time of surgery; (2) tumour histology, site/stage; (3) treatment, including neoadjuvant or adjuvant oncological treatment and surgical approach; and (4) postoperative complications classified according to the Clavien-Dindo scoring system (CDS).[16]

### Exposures
The following three types of surgery were used as exposures: (1) open transthoracic or transhiatal oesophagectomy (OE; reference group), (2) MIE (thoracoscopic-laparoscopic) and (3) HMIE (thoracoscopic/open abdomen or laparoscopic/open chest). Data on surgical technique were obtained from the medical records.

### Outcomes
The primary outcome was malnutrition measured by the abbreviated Patient-Generated Subjective Global Assessment (abPG-SGA) questionnaire (online supplemental figure 1) that was completed by the patient at one time point 1 year after surgery. This is a screening tool to detect patients at risk of malnutrition and is recommended for use in oncology patients.[17 18] The abPG-SGA is an abbreviated version of the PG-SGA questionnaire and has been found valid to detect malnutrition when used in an outpatient oncology clinic.[19]

The abPG-SGA contains four sections which concern body weight, food intake, symptoms reducing food intake and activities/function. The first section about body weight asks for the patient's current weight and height, weight at 1 and 6 months ago, and if the patient's weight has 'decreased'/'increased'/'has been stable' during the last 2 weeks. The second section concerns food intake where the patient compares his/her food intake to their normal intake as 'unchanged'/'more than usual'/'less than usual'. The patient is then asked to relate his/her food intake to the habitual food intake prior to cancer symptoms/diagnosis (in the original questionnaire patients related food intake to 1 month ago). If the answer was 'less than usual' the patient responded to the follow-up question: What type of food do you eat? with the following response alternatives 'normal food but less than normal amount'/'little solid food'/'only liquids'/'only nutritional supplements'/'very little of anything'/'only tube feeding or parenteral nutrition'. Section three covers symptoms that have kept the patient from eating enough during the past 2 weeks (several answers possible): 'no problem'/'no appetite, just did not feel like eating'/'nausea'/'vomiting'/'constipation'/ 'diarrhoea'/'mouth sores'/'dry mouth'/'things taste funny or have no taste'/'smells bother me'/'problems swallowing'/'feel full quickly'/'pain'/'other'. The forth section concerns activities/function and asks the patients to rate their general activity over the past months: 'normal'/'not my normal self, but have fairly normal activities'/'not feeling up to things, in bed/chair half the day'/'able to be little active and spend most of the day in bed/chair'/'rarely out of bed'. The score from each part totals up a sum that determines the patient's nutritional status. A higher score indicates more severe malnutrition.[17 19]

The secondary outcomes were postoperative weight loss between (1) time of surgery and 6 months postoperatively and (2) time of surgery and 1 year after surgery. The weight at surgery was collected from medical records, weight at 6 months was reported by the patient in the abPG-SGA questionnaire, and weight at 1 year was measured by the research nurse. Weight loss was calculated by comparing weight measures at surgery and weight at 6 months and at 1 year.

### Statistical analysis
Logistic regression models were used to analyse the associations between surgical techniques (OE (reference group), MIE, HMIE) and malnutrition (yes, no) and weight loss at 6 months and at 1 year postoperatively. Weight loss was categorised as (1) no weight loss/weight gain, (2) ≤10% wt loss, (3) >10% wt loss. Results were presented as ORs with 95% CIs, adjusted for potential confounders: age (longitudinal), sex (male, female), Charlson's Comorbidity Index (0, 1, ≥2), preoperative body mass index (BMI) (<25, ≥25), pathological tumour stage (0–I, II, III–IV), neoadjuvant therapy (yes, no), enteral/parenteral nutrition support during at least 1 week postoperatively (yes, no) and postoperative complications (CDS; low grade (0–II), high grade (III–IV)). Furthermore, an explanatory analysis was conducted to further adjust for anastomotic leak defined as clinically significant or radiologically

confirmed anastomotic leak (yes, no), intensive care unit stay (continuous) and hospital stay (continuous). Weight loss was also presented as median weight loss percentage, combined with the IQR for each group. In the analyses of the abPG-SGA questionnaire, a summarised score of >6 was considered as malnutrition. For body weight, food intake, symptoms reducing food intake and activities/function a score of >2 was considered as a lower/worse condition in each group, with a reference score of <2. A $p < 0.05$ was considered statistically significant.

## Patient and public involvement

The Surgical Care Science research partnership group including patients with oesophageal cancer were involved in the planning of OSCAR cohort study. The partnership group patients were involved in discussions around the study questions, as well as on the potential outcome measures collected during the study. The partnership group were not involved in recruitment of participants, but have been involved in the dissemination of research findings and increasing public awareness of oesophageal cancer. Regarding this study, the research partnership group patients suggested nutritional as an important outcome requiring more studies. Findings from this study will be communicated to patients and healthcare professionals through conferences and patient meetings.

## RESULTS

### Participants

Between January 2013 and April 2018, 675 patients underwent oesophageal cancer surgery in Sweden. Of these, 511 (76%) survived for at least 1 year, 85 were not reachable and 2 were excluded because of cognitive impairment, leaving 424 patients eligible for inclusion. Of these, 281 (66%) patients completed the 1-year assessment and clinical data were available for 247 (58%) patients. In total, 239 patients were included in the analyses. A comparison of study population and a population-based nationwide study of patients undergoing oesophagectomy in Sweden during the same period to examine the potential selection bias suggested no major difference in distribution of different surgical approaches in this study compared with population (online supplemental table 1).

### Patient characteristics

Patients were evenly distributed in the three surgical groups (OE 36%, MIE 33%, and HMIE 31%) (table 1). The mean age was 66.3 (±8.7) years at time of surgery and most patients were male (87%). The distribution of preoperative BMI and weight loss were similar across the three groups. Preoperative neoadjuvant therapy was most often used in the HMIE group (84%), while a higher pathological tumour stage was more frequently seen in the MIE group (37%), compared with the other groups. Almost all patients had a gastric conduit, while few patients in the OE group had an oesophagojejunostomy. A neck anastomosis was more common in the MIE group, compared

with the OE and HMIE groups. Patients who underwent HMIE less often received enteral/parenteral nutrition for at least 1 week, compared with the other groups. Patients with jejunostomy received enteral feeding for 14 days in median.

### Nutritional status

One-third (30%) of all patients had malnutrition 1 year postoperatively, based on the abPG-SGA (table 2), and it was highest in the MIE group (42%), compared with the OE (25%) and the HMIE (22%) groups. The MIE group had the highest prevalence of reduced food intake (63% vs 45% after HMIE vs 39% after OE), symptoms reducing food intake (60% vs 45% after HMIE vs 60% after OE). The number of reported symptoms reducing food intake (each symptom counted once for each patient, one patient could have multiple symptoms) in the MIE group was higher in the MIE group (121), compared with HMIE (71) and OE (107). Furthermore, abnormal activities/function were more common after MIE (45% vs 32% after HMIE vs 43% after OE). In multivariable analysis, the point estimate for the risk of malnutrition was high after MIE, compared with OE ($OR_{adj}$ 1.86, 95% CI 0.91 to 3.82), but this difference was statistically non-significant. A statistically significant association was observed between MIE and reduced food intake at 1 year after surgery ($OR_{adj}$ 2.87, 95% CI 1.47 to 5.61), compared with OE (table 3). No associations were observed between surgical technique and symptoms reducing food intake or activities/function.

In a sensitivity analysis, the level of anastomosis was included in the model, but the results remained the same, except that the point estimate of reduced food intake after MIE was increased to 3.13 (95% CI 1.53 to 6.39). Furthermore, in an attempt to minimise the risk that patients in the OE group reconstructed with jejunal conduit affected the outcomes, these eight patients were excluded from the analyses, but the results remain unchanged. Lastly, adjustment for anastomotic leak, intensive care unit stay and hospital stay in the explanatory analysis did not change the point estimates compared with the main analysis (online supplemental table 2).

### Weight loss

The patients lost most weight during the first 6 months postoperatively in all groups and it was most pronounced after HMIE, compared with both OE and MIE. A similar pattern was observed for weight loss at 1 year after surgery (figure 1). Surgical technique was not associated with weight loss at 6 months or 1 year after surgery (table 3). Additional adjustment for anastomotic leak, intensive care unit stay and hospital stay did not change the estimates (online supplemental table 2).

## DISCUSSION

The results of this population-based nationwide cohort study suggest that patients operated with MIE and HMIE

**Table 1** Distribution of patient characteristics in relation to the surgical technique used for oesophagectomy for cancer in a Swedish population-based cohort study

| | All | Open oesophagectomy (OE) | Minimally invasive oesophagectomy (MIE) | Hybrid minimally invasive oesophagectomy (HMIE) |
|---|---|---|---|---|
| | No (%) | | | |
| No of patients | 239 (100) | 87 (36) | 78 (33) | 74 (31) |
| Sex (male) | 207 (87) | 69 (79) | 70 (90) | 68 (92) |
| At time of surgery | | | | |
| Age (mean±1 SD) | 66.3±8.7 | 65.8±8.4 | 66.2±9.8 | 66.9±7.9 |
| Preoperative BMI (mean+1SD) | | | | |
| <25 | 36 (15) | 15 (17) | 12 (15) | 9 (12) |
| ≥25 | 203 (63) | 72 (83) | 66 (85) | 65 (88) |
| Preoperative weight loss | | | | |
| No weight loss/weight gain | 23 (10) | 10 (11) | 4 (5) | 9 (12) |
| <10% | 79 (33) | 26 (30) | 32 (41) | 21 (28) |
| ≥10% | 137 (57) | 51 (59) | 42 (54) | 44 (59) |
| Preoperative neoadjuvant treatment | 188 (79) | 68 (78) | 58 (74) | 62 (84) |
| Pathological tumour stage | | | | |
| 0–I | 79 (33) | 32 (37) | 16 (21) | 31 (42) |
| II | 80 (33) | 26 (30) | 33 (42) | 21 (28) |
| III–IV | 80 (33) | 29 (33) | 29 (37) | 22 (30) |
| Organ substitute | | | | |
| Gastric tube | 219 (92) | 73 (84) | 74 (95) | 72 (97) |
| Esophagojejunostomy | 10 (4) | 8 (9) | 1 (1) | 1 (1) |
| Colonic interposition | 2 (1) | 1 (1) | 1 (1) | 0 (0) |
| Level of anastomosis | | | | |
| Neck | 38 (16) | 9 (10) | 20 (26) | 9 (12) |
| Thorax | 187 (78) | 70 (80) | 54 (69) | 63 (85) |
| Abdomen | 6 (3) | 5 (6) | 0 (0) | 1 (1) |
| Charlson's Comorbidity Index | | | | |
| 0 | 116 (54) | 43 (49) | 37 (47) | 36 (49) |
| 1 | 77 (28) | 27 (31) | 23 (29) | 27 (36) |
| ≥2 | 46 (18) | 17 (20) | 18 (23) | 11 (15) |
| Enteral/parenteral nutrition | 175 (73) | 68 (78) | 60 (77) | 47 (64) |
| Postoperative complications | | | | |
| Low grade (CDS 0–II) | 144 (60) | 51 (59) | 44 (56) | 49 (66) |
| High grade (CDS III–IV) | 95 (40) | 36 (41) | 34 (44) | 25 (34) |
| Anastomotic leak | 41 (17) | 9 (10) | 21 (27) | 11 (15) |
| ICU stay (median, (IQR)) | 2 (1–5) | 3 (1–6) | 3 (1–6) | 2 (1–3) |
| Hospital stay (median, (IQR)) | 16 (12–25) | 16 (13–26) | 16.5 (12–29) | 15 (11–20) |

BMI, body mass index; CDS, Clavien-Dindo Score; ICU, intensive care unit.

had no increased risk in malnutrition compared with OE, but those operated with MIE were more likely to have reduced food intake 1 year postoperatively, compared with OE. No association between surgical technique and weight loss up to 1 year postoperatively was present.

The strength of this study is the prospective and population-based study design with longitudinal collection of data. Clinical data at time of surgery were collected from medical records and reviewed according to a predefined study protocol to ensure consistency

**Table 2** Results of the abPG-SGA questionnaire completed 1 year after oesophagectomy for cancer in relation to surgical technique in a Swedish population-based cohort

| abPG-SGA | All | Open oesophagectomy | Minimally invasive oesophagectomy | Hybrid minimally invasive oesophagectomy |
|---|---|---|---|---|
| | | | No (%) | |
| No of patients | 239 | 87 | 78 | 74 |
| Body weight | | | | |
| Reduced | 25 (10) | 11 (13) | 7 (9) | 7 (9) |
| Not changed | 173 (72) | 64 (74) | 51 (65) | 58 (78) |
| Increased | 39 (16) | 10 (11) | 20 (26) | 9 (12) |
| Food intake | | | | |
| Unchanged | 111 (46) | 46 (53) | 27 (35) | 38 (51) |
| More than usual | 8 (3) | 5 (6) | 1 (1) | 2 (3) |
| Less than usual | 116 (49) | 34 (39) | 49 (63) | 33 (45) |
| If less than usual, I am now taking: | | | | |
| Normal food but less amount | 108 (45) | 31 (36) | 44 (56) | 33 (45) |
| Little solid food | 3 (1) | 0 (0) | 3 (4) | 0 (0) |
| Only liquids/nutritional supplements | 1 (0) | 0 (0) | 1 (1) | 0 (0) |
| Very little of everything | 1 (0) | 1 (1) | 0 (0) | 0 (0) |
| Only tube feedings/nutritional by vein | 3 (1) | 2 (2) | 1 (1) | 0 (0) |
| Symptoms* | | | | |
| No problem eating | 107 (45) | 35 (40) | 31 (40) | 41 (55) |
| No appetite | 44 (18) | 15 (17) | 20 (26) | 9 (12) |
| Nausea | 34 (14) | 15 (17) | 14 (18) | 5 (7) |
| Constipation | 7 (3) | 3 (3) | 3 (4) | 1 (1) |
| Mouth sores | 3 (1) | 0 (0) | 1 (1) | 2 (3) |
| Things tast funny or have no taste | 19 (8) | 6 (7) | 9 (12) | 4 (5) |
| Problems swallowing | 35 (15) | 16 (18) | 13 (17) | 6 (8) |
| Pain† | 14 (6) | 4 (5) | 5 (6) | 5 (7) |
| Other‡ | 13 (5) | 5 (6) | 5 (6) | 3 (4) |
| Vomiting | 14 (6) | 4 (5) | 7 (9) | 3 (4) |
| Diarrhoea | 20 (8) | 8 (9) | 8 (10) | 4 (5) |
| Dry mouth | 11 (5) | 4 (5) | 2 (3) | 5 (7) |
| Smells bother me | 8 (3) | 2 (2) | 5 (6) | 1 (1) |
| Feel full quickly | 77 (32) | 25 (29) | 29 (37) | 23 (31) |
| Total number of symptoms reported: | 299 | 107 | 121 | 71 |
| Activities/function | | | | |
| Normal with no limitations | 143 (60) | 50 (57) | 43 (55) | 50 (68) |
| Not my normal self | 70 (29) | 28 (32) | 21 (27) | 21 (28) |
| Not feeling up to most things | 17 (7) | 6 (7) | 8 (10) | 3 (4) |
| Able to be little active | 6 (3) | 1 (1) | 5 (6) | 0 (0) |
| Pretty much bed ridden | 1 (0) | 1 (1) | 0 (0) | 0 (0) |
| Patients with malnutrition (total sum score) | 71 (30) | 22 (25) | 33 (42) | 16 (22) |

*Several answers possible. (Fatigue not included in the Swedish version of the abPG-SGA[17]).
†No patient gave information about where they had pain.
‡For example, throat irritation, the food does not taste like before, feeling of strangulation in end of meal, tooth problems, dumping, concern because of family problems, swelling in throat.
abPG-SGA, abbreviated Patient-Generated Subjective Global Assessment.

**Table 3** Associations between surgical technique for oesophagectomy due to cancer and nutritional status and weight loss 1 year after surgery in a Swedish population-based cohort study

| | Open oesophagectomy | Minimally invasive oesophagectomy | Hybrid minimally invasive oesophagectomy |
|---|---|---|---|
| | (Reference) | OR (95% CI)* | OR (95% CI)* |
| Malnutrition—total score from abPG-SGA | 1.0 | 1.86 (0.91 to 3.82) | 0.92 (0.42 to 2.04) |
| Reduced food intake | 1.0 | 2.87 (1.47 to 5.61) | 1.32 (0.68 to 2.59) |
| Symptoms reducing food intake† | 1.0 | 1.32 (0.67 to 2.61) | 0.93 (0.46 to 1.90) |
| Decreased activities/function | 1.0 | 1.81 (0.65 to 5.02) | 0.50 (0.12 to 2.11) |
| Weight | | | |
| Weight loss 6 months after surgery | | | |
| ≤10% | 1.0 | 1.26 (0.38 to 4.19) | 1.14 (0.32 to 4.02) |
| >10% | 1.0 | 1.51 (0.47 to 4.90) | 1.59 (0.47 to 5.40) |
| Weight loss 1 year after surgery | | | |
| ≤10% | 1.0 | 3.08 (0.86 to 10.95) | 0.90 (0.31 to 2.61) |
| >10% | 1.0 | 2.06 (0.60 to 7.04) | 0.96 (0.36 to 2.57) |

*Adjusted for age, sex, Charlson's Comorbidity Index, preoperative BMI, pathological tumour stage, neoadjuvant therapy, enteral/parenteral nutrition support and postoperative complications.
†Nausea, diarrhoea, dry mouth, problems swallowing, feel full quickly, fatigue, pain, etc.
abPG-SGA, abbreviated Patient-Generated Subjective Global Assessment; BMI, body mass index.

and uniformity of the data. The cohort had relatively high inclusion rate, and completeness of the abPG-SGA questionnaire in survivors 1 year postoperatively was very high. The abPG-SGA questionnaire is a validated and commonly used instrument that has been reported to be a valid tool to measure malnutrition in oncology patients.[19] The results may be generalisable to hospitals in Western countries where these surgical techniques are used. Missing data are in general low when collected by a research nurse in patients' homes. Weaknesess of this study include that all patients who died during 1 year after oesophagectomy were not eligible for recruitment in the

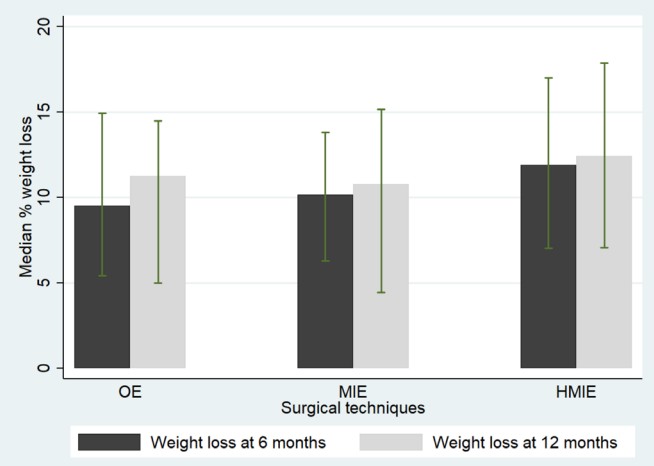

**Figure 1** Median (%) weight loss at 6 months and at 1 year after surgery in patients who underwent oesophagectomy with different surgical techniques. OE, open oesophagectomy; MIE, minimally invasive oesophagectomy; HMIE, hybrid minimally invasive oesophagectomy.

study, and that there is potential selection bias at the inclusion time of 1 year after surgery despite the distributions of surgical approaches being similar in this study and the population. Moreover, the patients' self-reported weight measure could be biased, although this is assumed to be a minor problem since keeping track of weight is a central concern among these patients. In the section about food intake in the abPG-SGA, the nurse asked these patients to relate their actual food intake to their habitual eating prior to diagnosis, instead of 1 month ago, to make it more suitable for this patient group. This may give a more correct picture of the nutritional status in this group since comparing to 1 month ago would not have shown any difference because their eating problems started at time of surgery or earlier.

Previous studies comparing malnutrition between MIE and OE are sparse, as very few have looked at malnutrition in the long term, and none using the abPG-SGA questionnaire. The previous studies assessing malnutrition have used weight loss or BMI as a proxy for nutritional status[20 21] or laboratory values.[6 22] In an Italian case–control study, serum albumin levels were higher at days 1, 3 and 8 in patients undergoing HMIE, compared with OE.[22] An American study suggested earlier oral intake in patients who underwent MIE, compared with OE.[6] The results from the current study suggest that MIE has a greater negative effect on the patients' food intake, compared with OE, which is the opposite of what we hypothesised a priori. However, there was no significant difference in malnutrition, or other nutritional symptoms between the surgical techniques. Based on previous research, MIE should lead to less postoperative complications and shorter length of hospital stay, compared with

OE.[5–7] In this study, major complications were similar between the groups and likely not a relevant factor in food intake symptoms. Patients undergoing MIE had higher occurrence of anastomotic leaks (27%) than those operated with OE (10%) or HMIE (15%). The higher percentages of anastomotic leaks are most likely related to the ongoing learning curve of MIE and HMIE in Sweden during the study period. A previous dutch study suggested that there are more anastomotic leaks (19%) during the learning phase of MIE compared with after completed learning (5%).[23] Therefore, it could be speculated that anastomotic leaks related to MIE and HMIE would explain the observed differences in food intake. However, an additional adjustment for anastomotic leak, intensive care unit stay and hospital stay in the explanatory analysis did not result in major changes in the point estimates compared with the main analysis. Unfortunately, no data on the management of anastomotic leaks were available in this study, and the possible effects of sequelae of anastomotic leaks could not be furhter explored. On the other hand, neck anastomosis is known to be associated with more anastomotic strictures, compared with chest anastomosis, which might affect the food intake.[24] As the MIE group had more neck anastomoses than HMIE and OE groups, a sensitivity analysis adjusted for anastomotic location was conducted but the results were no different from the main analysis. The conduit used for reconstruction might also affect food intake, but excluding jejunal interposition grafts did not change the results. Anastomotic techniques (ie, handsewn vs circular stapled vs linear stapled) might also differ between MIE, HMIE and OE. For example, end-to-side stapled and hand-sewn, as well as linear stapled side-to-side anastomoses have been described in Sweden.[25 26] However, these differences remain speculative, as these data were not available, and further studies are needed. Lastly, early oral feeding is central to any Enhanced Recovery After Surgery (ERAS) programmes,[27] and could impact the long-term nutritional recovery of these patients. However, participation in ERAS programmes or the nutritional protocols in the early postoperative period were not available in the dataset. On the other hand, early oral nutrition was not associated to weight loss compared with jejunostomy after 1-month past surgery in a Dutch cohort study of 114 MIE patients.[28] As oesophageal cancer strongly affects the patients' food intake before and after surgery, the associated weight loss is often substantial.[29] Some previous studies have investigated how surgical technique affects patients' weight in the long-term. A Swedish clinical study observed no difference in weight loss during the first postoperative year in patients who underwent MIE, compared with OE.[20] Our results suggested a higher weight loss in the HMIE group, both at 6 months and 1 year postoperatively, but this difference was non-significant in the adjusted analyses. In addition to no nutritional differences between the surgical methods, the results could be explained by too small number of patients in the analysis. However, the weight loss in the OE group was increasing up to 1 year after surgery, and studies with longer-term follow-up might be warranted.

Even though there were no major differences in malnutrition between MIE and open oesophagectomy, a striking finding of this study is that one-third (30%) of all included patients were malnourished at 1 year after surgery. This highlights the importance of nutritional support and long-term follow-up from the healthcare for patients who undergo oesophagectomy. Since oesophagectomy results in a permanent anatomical change of the upper gastrointestinal tract, besides nutritional strategies in the immediate postoperative phase such as early enteral feeding, nutritional interventions for postoperative complications, attention to long-term nutritional intake and status are also of high importance.[29] While surgical technique might not be of high importance, the impact of anastomotic technique on nutrition in patients with oesophageal cancer should be further evaluated. With improved knowledge, the utility of long-term dietitian support in different patient groups could be clarified.

In conclusion, this population-based prospective cohort study suggests that while there was no increased risk of malnutrition between minimally invasive compared with OE, MIE is associated to reduced food intake 1 year postoperatively. No significant differences were present between surgical technique and weight loss. Further studies on nutritional impact of surgical techniques in oesophageal cancer are needed.

**Acknowledgements** We thank all study participants and the members of the Surgical Care Science patient research partnership group for their contribution to the OSCAR study.

**Contributors** JHK, HR, FK, PA and PL contributed with input in designing the study. Data collection and assembly of data were performed by KM and PL. Data analysis was conducted by AJ. All authors (JHK, HR, FK, AJ, PA, KM and PL) contributed in interpreting the results, manuscript writing and approval of the final version of the manuscript. PL is the guarantor.

**Funding** We also would like to thank the Cancerfonden (grant number 180685), the Cancer Research Funds of Radiumhemmet (grant number 171103) and the Stockholm County Council (grant number LS 2018-1157) for financial support. Pernilla Lagergren is supported by the NIHR Imperial Biomedical Research Centre (BRC) for her position at Imperial College London, London, UK.

**Disclaimer** The funding sources had no role in the design and conduct of the data collection, management, analysis and interpretation of the data or preparation review or approval of the manuscript.

**Competing interests** None declared.

**Patient and public involvement** Patients and/or the public were involved in the design, or conduct, or reporting, or dissemination plans of this research. Refer to the Methods section for further details.

**Patient consent for publication** Not applicable.

**Ethics approval** This study involves human participants and was approved by Regional Ethical Review Board in Stockholm, Sweden, diary number 2013/844-31/1. Participants gave informed consent to participate in the study before taking part.

**Provenance and peer review** Not commissioned; externally peer reviewed.

**Data availability statement** All data underlying the findings are presented in the manuscript. The datasets generated and/or analysed in the current study will not be publicly available due to the ethical review act but will be available from the principal investigators of the study on reasonable request.

**ORCID iDs**
Joonas H Kauppila http://orcid.org/0000-0001-6740-3726
Poorna Anandavadivelan http://orcid.org/0000-0002-3471-1578

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
