## [Reviewer comments · BMJ Open]

ARTICLE DETAILS

TITLE (PROVISIONAL)	Minimally invasive surgical techniques for esophageal cancer and nutritional recovery – a prospective population-based cohort study.
AUTHORS	Kauppila, Joonas; Rosenlund, Helen; Klevebro, Fredrik; Johar, Asif; Anandavadivelan, Poorna; Mälberg, Kalle; Lagergren, Pernilla

VERSION 1 – REVIEW

REVIEWER	Nieuwenhuijzen, Grard Catharina Hospital Eindhoven, Department of Surgery
REVIEW RETURNED	12-Dec-2021

GENERAL COMMENTS	This population-based study has compared nutritional status, malnutrition and weight loss between total minimal invasive (MIE), hybrid minimal invasive (HMIE) and Open Esophagectomy (OE). 675 Patients were operated upon between 2013 and 2018, 424 patients were alive 1 year after surgery of which 281 completed the assessment and of which eventually 239 patients were included. The study has well been performed, with adequate statistical methods and the manuscript is well written. While actually surgical reconstructive techniques in MIE and open surgery do not differ that much (only in MIE more neck anastomosis), the results are remarkable: MIE was associated with reduced food intake 1 year after surgery, however no differences in weight loss. Because the results are actually quite unexpected, I am wondering whether or not they have included all relevant factors in their analyses. Hence, I have some major concerns regarding some aspects of this study. 1. First the selection of patients: was the selection/inclusion of patients evenly distributed between all three groups regarding drop out due to death and inclusion? If there is an uneven distribution then it should be statistically corrected.2. Second, anastomotic leakage (AL) has been shown to be highly associated with a prolonged stay in hospital, reduced food intake due to general weakness and anastomotic strictures. Between 2013 and 2018 MIE was still in a learning curve in Sweden and high AL incidences have been described in the early phases. In their study they have shown that there was no difference in high grade complications but I would be interested in AL specifically. So please could they provide these numbers and include them in the multivariable analysis (MVA) if there is a univariable significant difference between the incidence and grade of AL.
--

	3. Third, the same for ICU and hospital stay. A long ICU and hospital stay leads to sarcopenia and fragility, leading to an inability to recover fast after surgery and energy to achieve a normal food intake. So please could they provide these numbers and include them in the multivariable analysis (MVA) if there is a univariable significant
--	---

REVIEWER	Bollschweiler, E. University of Cologne
REVIEW RETURNED	19-Dec-2021

GENERAL COMMENTS	This is a very important study. The results are surprising. Have the authors ideas about the reasons of the differences in the main outcome? I have no additional comments.
--

VERSION 1 – AUTHOR RESPONSE

Reviewer: 1

Dr. Grard Nieuwenhuijzen, Catharina Hospital Eindhoven

Comments to the Author:

This population-based study has compared nutritional status, malnutrition and weight loss between total minimal invasive (MIE), hybrid minimal invasive (HMIE) and Open Esophagectomy (OE). 675 Patients were operated upon between 2013 and 2018, 424 patients were alive 1 year after surgery of which 281 completed the assessment and of which eventually 239 patients were included.

The study has well been performed, with adequate statistical methods and the manuscript is well written.

While actually surgical reconstructive techniques in MIE and open surgery do not differ that much (only in MIE more neck anastomosis), the results are remarkable: MIE was associated with reduced food intake 1 year after surgery, however no differences in weight loss. Because the results are actually quite unexpected, I am wondering whether or not they have included all relevant factors in their analyses. Hence, I have some major concerns regarding some aspects of this study.

Q1. First the selection of patients: was the selection/inclusion of patients evenly distributed between all three groups regarding drop out due to death and inclusion? If there is an uneven distribution then it should be statistically corrected.

Authors' response: We thank for these important remarks regarding the validity of the study. Only patients surviving 1 year after esophagectomy were included. Therefore, the study findings are applicable to only patients surviving at least one year. For the study outcomes, the death prior to 1 year of follow-up (patient inclusion) is not relevant, as nutrition cannot be studied in patients that are already dead, and thus should not lead to selection bias.

We agree that selection could have potentially happened during patient recruitment and selection. Using another population-based study including >98% of all esophagectomy patients in Sweden, we were able to compare the proportions of different surgical approaches between the present study and the population. Unfortunately, a direct comparison of included and non-included patient was not possible due to legislative issues related to study permits. Based on comparisons of

patients included in the population-based SESS-study, and the present OSCAR study (below), there were no major differences in proportions of surgical approaches between the two studies.

	OSCAR-study	SESS-study
Surgical approach		
Minimally invasive	28 (21.7%)	76 (14.9%)
Hybrid minimally invasive	37 (28.6%)	133 (27.6%)
Open surgery	64 (49.6%)	273 (56.6%)

In a year-by-year comparison, there was slightly larger proportion of minimally invasive surgery patients in OSCAR study in 2015, but there was no statistically significant difference in proportions between the studies for those recruited during year 2015 either (p=0.13).

	2013		2014		2015	
	OSCAR-study	SESS-study	OSCAR-study	SESS-study	OSCAR-study	SESS-study
Surgical approach						
Minimally invasive	2 (5.3%)	13 (8.3%)	8 (19.1%)	25 (14.6%)	18 (36.7%)	38 (24.5%)
Hybrid minimally invasive	3 (7.9%)	11 (7.1%)	17 (40.5%)	72 (42.1%)	17 (34.7%)	50 (32.3%)
Open surgery	33 (86.8%)	132 (84.6%)	17 (40.5%)	74 (43.3%)	14 (28.6%)	67 (43.2%)

Revisions: The proportions of minimally invasive (total and hybrid combined) or open esophagectomy are presented in Results section and supplementary table 1 for patients included in the present study and the Swedish population during the same time period. We added a comment on selection bias being a potential weakness of the study in the discussion section

Q2. Second, anastomotic leakage (AL) has been shown to be highly associated with a prolonged stay in hospital, reduced food intake due to general weakness and anastomotic strictures. Between 2013 and 2018 MIE was still in a learning curve in Sweden and high AL incidences have been described in the early phases. In their study they have shown that there was no difference in high grade complications but I would be interested in AL specifically. So please could they provide these numbers and include them in the multivariable analysis (MVA) if there is a univariable significant difference between the incidence and grade of AL.

Authors' response: We agree that anastomotic leaks could affect the nutritional aspects in these patients, and it may explain the results to some extent. On the other hand, AL is also on the causal pathway between surgical method and nutrition, which is why it was left out from the previous analysis. The data on the severity of anastomotic leaks was not specifically collected for this study and anastomotic leaks were categorized as yes, or no with the definition "clinically significant or radiologically confirmed anastomotic leak". With these limitations, we decided to add an explanatory model which includes anastomotic leak (and ICU and hospital stay) as a covariates into the supplement. This analysis did not differ in any aspect from the main analysis (table below)

Supplementary Table 2. Explanatory analysis of anastomotic leak and intensive care unit and hospital stay on associations between surgical technique for esophagectomy due to cancer and nutritional status and weight loss 1 year after surgery in a Swedish population-based cohort study

	Open esophagectomy (Reference)	Minimally invasive esophagectomy OR (95% CI) ¹	Hybrid minimally invasive esophagectomy OR (95% CI) ¹
Malnutrition – total score from abPG-SGA	1.0	1.80 (0.84-3.86)	0.90 (0.39-2.07)
Reduced food intake	1.0	3.25 (1.60-6.57)	1.46 (0.73-2.93)
Symptoms reducing food intake ²	1.0	1.35 (0.66-2.77)	0.96 (0.46-2.02)
Decreased activities/function	1.0	1.44 (0.48-4.35)	0.40 (0.09-1.77)
Weight			
Weight loss 6 months after surgery			
≤10%	1.0	1.14 (0.31-4.12)	1.04 (0.28-3.83)
>10%	1.0	1.49 (0.42-5.30)	1.56 (0.44-5.55)
Weight loss 1 year after surgery			
≤10%	1.0	2.86 (0.74-11.11)	0.73 (0.23-2.38)
>10%	1.0	1.56 (0.42-5.77)	0.84 (0.28-2.51)

¹ Adjusted for age, sex, Charlson's comorbidity index, preoperative BMI, pathological tumor stage, neoadjuvant therapy, enteral/parenteral nutrition support, postoperative complications, anastomotic leak, intensive care unit stay, and hospital stay.

² Nausea, diarrhoea, dry mouth, problems swallowing, feel full quickly, fatigue, pain etc. Abbreviations: OR, Odds ratio; CI, confidence interval.

Revisions: The above table was added to the supplement. Furthermore, methods, results, and discussion sections were revised to note these analyses.

Q3. Third, the same for ICU and hospital stay. A long ICU and hospital stay leads to sarcopenia and fragility, leading to an inability to recover fast after surgery and energy to achieve a normal food intake. So please could they provide these numbers and include them in the multivariable analysis (MVA) if there is a univariable significant difference between the ICU and hospital stay.

Authors' response: We thank for this notification. It is well known that hospital stay (and potentially ICU stay) are strongly affected by the exposure i.e. minimally invasive or open esophagectomy as shown in all RCTs conducted on the topic. However, the length of stay should be shorter in patients undergoing MIE than those undergoing open esophagectomy. Anyway, this adjustment will to some extent adjust for the exposure itself. For these reasons, we have done an exploratory analysis including hospital and ICU stay (and anastomotic leak) as covariates and added this analysis in the supplement. See answer to Q2 for details.

Reviewer: 2

Q1: This is a very important study. The results are surprising. Have the authors ideas about the reasons of the differences in the main outcome?

I have no additional comments.

Authors' response: We agree that the results are surprising, as our thought was that the methods would be either equal, or favor MIE. It is hard to say why there are differences, but one can always speculate on the reasons behind the findings. The methods of anastomosis could be different between open and minimally invasive surgery, with potentially more hand-sewn anastomoses in the open group, as well as different staplers in different type procedures. Due to lack of certain data, we think that repeating the study in another setting, as well as exploring the aforementioned factors would help in defining whether these associations are real and for which reasons.

VERSION 2 – REVIEW

REVIEWER	Nieuwenhuijzen, Grard Catharina Hospital Eindhoven, Department of Surgery
REVIEW RETURNED	04-Apr-2022

GENERAL COMMENTS	I want to thank the authors for their extensive answers and additional analyses. However, I still have some major concerns which I would like to address. 1. The issue of a possible selection is only partly addressed. I agree that they could only evaluate the patients that had survived more than 1 year, since nutrition cannot be studied in patients that have already died. However, data on the 1-year survival in the three groups have still not been shown in the results section. Hence, I would still suggest to mention 1 year mortality in the results and address these data in the discussion. 2. The issue of AL has been addressed adequately in the additional explanatory analyses. However, AL has only been addressed as “yes or no” and we are not informed on the sequelae of AL which could have influenced and mediated the results of inadequate food intake significantly i.e., delay of food intake as a result of the healing time and stenosis. Especially if you observe the percentages of AL. In the MIE group 51% have experienced AL!! This is a very high percentage (much higher than shown in the literature), it is significantly higher than in the other groups (OE 22% and HMIE 27%). The sequelae of AL could have been the main reason of the observed reduced food intake. I agree that AL is associated with the surgical method, however, these percentages are not in concordance with other studies. It thus shows that MIE was in its learning curve in Sweden. Hence, the observed reduced food intake in MIE could have been mediated by the learning curve, AL and not by the surgical technique itself. If MIE would have been passed the learning curve and AL percentages would have been comparable with OE and HMIE (as shown in other studies) it remains to be awaited whether the observed phenomena would still be present. This high AL percentage, the possible learning curve, the missing sequelae of AL should be addressed (compared with the literature on MIE)
--

	much more extensively as a weakness of this study in the discussion. Actually, the differences in the use in cervical anastomosis (10%, 26% and 12% in the use of cervical anastomoses in OE, MIE and HMIE respectively) and AL (22%, 51% and 27% respectively) are that different between the groups and theoretically predominantly determining the nutritional observed phenomena, that I am in doubt whether the statistical methods used are the most appropriate. In fact, I am not a statistician, however I am wondering whether a Propensity Score Matching would not be more appropriate since linearity cannot be assumed and Some factors are mediators of the observed effect. Hence, I would propose to ask statistical expertise for their opinion. 3. This study confirms what has been shown in other studies that the weight loss and reduced food-intake in the early postoperative period is not related to long-term outcome. Hence the relevance of I sufficient food intake in the early postop period should be questioned. The long-term effects of early oral feeding following minimal invasive esophagectomy. (Berkelmans GHK et al. Dis Esophagus. 2018 Jan 1;31(1):1-8). Unfortunately, we are not informed on the nutritional protocols in the early period. Especially the indication for of supplements and tube feeding Hence, after 6 months the data could be flawed by the nutritional protocols and thus should be mentioned in the weaknesses of the study. 4. In conclusion, I am not convinced that the observed phenomena are caused by the surgical technique and are more mediated by other mediators such as AL, location of anastomosis and learning curves.
--	---

VERSION 2 – AUTHOR RESPONSE

Reviewer: 1

Dr. Grard Nieuwenhuijzen, Catharina Hospital Eindhoven

Comments to the Author:

I want to thank the authors for their extensive answers and additional analyses. However, I still have some major concerns which I would like to address.

Q1. The issue of a possible selection is only partly addressed. I agree that they could only evaluate the patients that had survived more than 1 year, since nutrition cannot be studied in patients that have already died. However, data on the 1-year survival in the three groups have still not been shown in the results section. Hence, I would still suggest to mention 1 year mortality in the results and address these data in the discussion.

Authors' response: We thank for the insightful previous and new comments that have arisen after the revisions. We agree that it would be interesting to mention 1-year mortality in the manuscript, and it might add some detail to the study. However, as the study recruitment was conducted prospectively

one year after surgery, i.e. only those patients who had already survived 1 year could give their informed consent, we unfortunately do not have this information, and it is not possible for us to retrieve 1-year mortality rate for these patients.

Revisions: We revised the study design section on page 5 to more clearly indicate that one-year survivors were recruited. Furthermore, we added this information to the discussion under weaknesses of the study as well (page 16).

Q2. The issue of AL has been addressed adequately in the additional explanatory analyses. However, AL has only been addressed as “yes or no” and we are not informed on the sequelae of AL which could have influenced and mediated the results of inadequate food intake significantly i.e., delay of food intake as a result of the healing time and stenosis. Especially if you observe the percentages of AL. In the MIE group 51% have experienced AL!! This is a very high percentage (much higher than shown in the literature), it is significantly higher than in the other groups (OE 22% and HMIE 27%). The sequelae of AL could have been the main reason of the observed reduced food intake. I agree that AL is associated with the surgical method, however, these percentages are not in concordance with other studies. It thus shows that MIE was in its learning curve in Sweden. Hence, the observed reduced food intake in MIE could have been mediated by the learning curve, AL and not by the surgical technique itself. If MIE would have been passed the learning curve and AL percentages would have been comparable with OE and HMIE (as shown in other studies) it remains to be awaited whether the observed phenomena would still be present. This high AL percentage, the possible learning curve, the missing sequelae of AL should be addressed (compared with the literature on MIE) much more extensively as a weakness of this study in the discussion. Actually, the differences in the use in cervical anastomosis (10%, 26% and 12% in the use of cervical anastomoses in OE, MIE and HMIE respectively) and AL (22%, 51% and 27% respectively) are that different between the groups and theoretically predominantly determining the nutritional observed phenomena, that I am in doubt whether the statistical methods used are the most appropriate. In fact, I am not a statistician, however I am wondering whether a Propensity Score Matching would not be more appropriate since linearity cannot be assumed and Some factors are mediators of the observed effect. Hence, I would propose to ask statistical expertise for their opinion.

Authors' response: We apologize for the embarrassing typos in the percentages of anastomotic leak in the table, as 21/78 is 27%, not 51%. The correct percentages for anastomotic leaks were 10% for OE, 27% for MIE, and 15% for HMIE. We still agree that the incidence of anastomotic leak is higher in the MIE group, and probably is related partially to learning curves, and partially to cervical anastomoses known to associate with more leaks. However, adjustment for anastomotic location and anastomotic leaks did not change the point estimates. It could be still argued that the potentially different grade or the treatment of anastomotic leak might explain the observed differences. but then there should be more grade III-IV complications in the MIE group, and we would expect at least a minor change towards no difference between MIE and OE in the analysis adjusted for leaks and ICU/hospital stay. Opposing this hypothesis, the adjusted exploratory analysis suggested even greater difference between the two surgical methods.

We have discussed with the senior biostatisticians of our group about the optimal way to analyze the data while adjusting for potential confounders. They all favor the multivariable logistic regression approach because it saves statistical power and provides in most instances very similar

results as the propensity score matching approach. The issues related to propensity score matching have been extensively discussed in the literature (for example PMID: 26712591). Although in rare instances a balance achieved in the covariate distribution may decrease the variance of the estimators, discarding cases in the matching process will result in smaller sample size and lead to increased variance, and may actually lead to more imbalances between the groups. This could cause either false positive results, or losing of results that actually exist between the groups.

The number of patients in the study is already relatively small and we would not want to further reduce the statistical power. We consider logistic regression with adjustments for confounders the most suitable method for the present study, and therefore we wish to keep to the a priori study protocol and current statistical methods.

Revisions: The percentages have been corrected in table 1. We have added more discussion related to learning curve of esophagectomy, as well as the above mentioned points as suggested on pages 17-18 in the discussion.

Q3. This study confirms what has been shown in other studies that the weight loss and reduced food-intake in the early postoperative period is not related to long-term outcome. Hence the relevance of I sufficient food intake in the early postop period should be questioned. The long-term effects of early oral feeding following minimal invasive esophagectomy. (Berkelmans GHK et al. Dis Esophagus. 2018 Jan 1;31(1):1-8). Unfortunately, we are not informed on the nutritional protocols in the early period. Especially the indication for of supplements and tube feeding Hence, after 6 months the data could be flawed by the nutritional protocols and thus should be mentioned in the weaknesses of the study.

Authors' response: Many thanks for this important remark. We have added the lack of nutritional protocols as a potential problem and cited the suggested study in the discussion section related to weight loss on page 18.

Q3. In conclusion, I am not convinced that the observed phenomena are caused by the surgical technique and are more mediated by other mediators such as AL, location of anastomosis and learning curves.

Authors' response: We agree that there are several factors explaining the changes or the lack of thereof, as described in the manuscript. However, we would like to stress that we have not implied causality between MIE and nutrition in any way in the present study.